# The Role of Seminal Oxidative Stress in Recurrent Pregnancy Loss

**DOI:** 10.3390/antiox12030723

**Published:** 2023-03-15

**Authors:** Rhianna Davies, Channa N. Jayasena, Raj Rai, Suks Minhas

**Affiliations:** 1Department of Metabolism, Digestion and Reproduction, Imperial College London, London W12 0NN, UK; 2Department of Obstetrics and Gynaecology, St Mary’s Hospital, Imperial College NHS Trust, London W2 1NY, UK; 3Department of Urology, Charing Cross Hospital, Imperial College NHS Trust, London W6 8RF, UK

**Keywords:** recurrent pregnancy loss, sperm DNA damage, reactive oxygen species, ROS, antioxidants

## Abstract

Recurrent pregnancy loss is a distressing condition affecting 1–2% of couples. Traditionally investigations have focused on the female, however more recently researchers have started to explore the potential contribution of the male partner. Seminal reactive oxygen species have a physiological function in male reproduction but in excess are suspected to generate structural and functional damage to the sperm. Evidence is mounting to support an association between elevated seminal reaction oxygen species and recurrent pregnancy loss. Studies suggest that the rates of sperm DNA damage are higher in the male partners of women affected by recurrent pregnancy loss compared with unaffected men. However, the available pool of data is conflicting, and interpretation is limited by the recent change in nomenclature and the heterogeneity of study methodologies. Furthermore, investigation into the effects of oxidative stress on the epigenome show promise. The value of antioxidant therapy in the management of recurrent pregnancy loss currently remains unclear.

## 1. Introduction

Recurrent pregnancy loss (RPL; recurrent miscarriage) is a devastating condition affecting 1–2% of couples [1]. Historically described as the loss of 3 or more consecutive pregnancies before 20 weeks of gestation, the definition was amended in 2018 by the European Society of Human Reproduction and Embryology [1,2,3]. ESHRE defines RPL as 2 or more pregnancy losses before 24 weeks gestation, without a need for them to be consecutive [3]. Investigation into causes of RPL have traditionally focused on the women yet advanced paternal age as well as genetic and epigenetic aberrations derived from the male gamete are also proposed to play a role [2,4]. Embryonic chromosomal abnormalities account for 30–57% of subsequent losses in couples affected by RPL [4]. Furthermore, an unbalanced translocation, derived from either the male or female partner, accounts for 2–5% of all RPL [4]. However, in around 50% of cases of RPL no cause is found, thus precluding targeted therapies [1,2,4,5]. Increasingly, attention has turned to a male factor contribution to the aetiology of RPL [6,7]. Reactive oxygen species (ROS), generated by seminal leucocytes and abnormal spermatozoa, are potential inducers of sperm damage [5,7,8,9]. This review will summarise the currently available literature regarding the proposed association between seminal ROS and RPL.

## 2. Spermatic Function

Fertilisation and post-fertilisation embryonic events require a complex stepwise process of tightly regulated events. Upon release from the testes, spermatozoa travel through the epididymis and are bathed in secretions from the male reproductive glands [10]. The sperm interacts with these secretions; indeed, it is now understood that certain proteins are actively imported into the sperm from the surrounding fluid [11]. Upon ejaculation into the female reproductive tract, the sperm encounter bicarbonate in the female vagina; this induces enzymatic activation of the sperm leading to alteration of the lipid and glycoprotein composition of their plasma membrane. This process is termed capacitation and enhances the sperms’ motility and metabolic energy to allow passage to the oocyte and through the surrounding follicle cells to bind with the zona pellucida [10]. Certain spermatic proteins have been identified to be crucial to the ability to bind. Upon binding, the sperm undergoes the acrosome reaction, inducing enzymatic action to burrow through the zona pellucida and to expose proteins in the sperm plasma membrane to bind to the oocyte plasma membrane. Contrary to prior dogma, the sperm contributes more to the zygote (fertilised egg) than just its DNA [12]. The sperm centriole also enters the egg, before duplicating. It has a functional role in the first mitotic division. The paternal genome is demethylated within the first 24 h of embryo development, much sooner than the female genome. The sperm DNA provides epigenetic codes such as for DNA methylation and post-translational modifications of proteins and histones. The paternal gamete is known to play a crucial role in placentation [7]. This is demonstrated by observing mouse embryos with 2 paternal genomes where the embryo development is aberrant but placental formation is preserved [13]. The paternal genome is inactivated until 2 days after fertilization [14]. However, once active, a defective paternal genome can lead to inadequate development of the blastocyst, unequal cleavage, failure to implant into the endometrium of the female partner and miscarriage [14]. To date, 6238 different proteins have been identified as carried by the human paternal gamete [12]. Their exact role is yet to be fully elucidated but studies have identified several discrete proteins involved in fertilisation, pre-implantation development of the fertilised embryo and post-implantation events [12]. Ergo, integrity of the DNA and additional proteins carried by the sperm is critical for embryo development and success of an early pregnancy.

## 3. Seminal Redox Balance

The term ROS describes an unstable molecule capable of extracting an electron from another molecule to achieve a stable state. This latter molecule is now at risk of becoming an unstable ROS itself, precipitating a chain of ROS production [15,16]. Seminal ROS, generated within the male reproductive tract, has a physiological role in the male reproductive system. Small amounts of ROS are needed for sperm capacitation, motility and the acrosome reaction, as well as fertilisation of oocytes [8,17]. The main source of ROS generation in the seminal fluid is by seminal leucocytes and abnormal spermatozoa [8]. Excess residual cytoplasm is a type of sperm anomaly observed to generate considerable ROS [18]. ROS generation in cytoplasmic droplets is mediated by the enzyme glucose-6-phosphate-dehydrogenase (G6PD) via 2 distinct pathways: nicotinamide-adenine-dinucleotide-phosphate (NADPH) in the sperm plasma membrane and NADPH-dependent oxidoreductase in the mitochondria [19,20,21]. Indeed, the sperm plasma membrane and the mitochondria are 2 established sites of ROS generation within sperm [22]. In excess, seminal ROS can damage the sperm. ROS induces breakage of DNA strands and chromatin cross-linking via NADPH pathways [23]. A prospective cohort study found a statistically significant correlation between human spermatozoa rich in cytoplasmic droplets, elevated ROS and sperm DNA damage [23]. In-vitro studies of human sperm have identified that ROS induces peroxidation of lipids in the sperm membrane; this impairs the flexibility of the sperm and reduces their motility [24,25]. ROS can also damage the mitochondria which are required to provide the motile energy of the sperm [24,25]. The generation of ROS from defective mitochondria induces sperm and mitochondrial damage; this results in a cycle of ROS generation [21]. Polymorphonuclear leukocytes (PMN) represent 50–60% of all seminal leucocytes. Alongside macrophages, PNM leukocytes can generate ROS [26]. Leucocytes may be activated by infection and inflammation. Compared to their non-activated counterparts, activated leucocytes produce up to 100 times higher levels of ROS [27,28]. The generation of ROS is thought to be enhanced by various factors including alcohol use, smoking, obesity, aging, psychological stress, intense physical exercise, medical co-morbidities including diabetes, infection and environmental exposures [17,29,30,31,32,33,34,35] (Figure 1) Furthermore, several studies have shown an association between varicoceles and increased ROS [36].

To maintain a balance between oxidative and reductive action, reducing agents are also produced to reduce cellular injury [37]. These so-called anti-oxidants are both produced by the male accessory glands and consumed in the diet [38]. This system allows for the beneficial action of ROS in the male reproductive tract whilst maintaining oxidative stress at a low enough level to avoid damage to the sperm. Evidence is mounting that an oxidative stress predominant environment plays a causative role in male factor infertility [39,40,41,42,43]. Study has observed a negative correlation between seminal ROS levels and sperm morphology and motility [43]. Indeed, studies measuring ROS via various methods find that 30–88% of infertile men have elevated seminal ROS [40,41,42]. The term Male Oxidative Stress Infertility (MOSI) has been coined to explain male infertility in the setting of elevated ROS [44]. RPL is a clinically distinct entity, differing from infertility by virtue of successful conception but subsequent inability to carry the conceptus to viability. However, given the biological plausibility that mechanisms able to induce infertility could also induce early pregnancy loss, investigators have designed studies to establish a potential link between elevated seminal ROS and RPL.

## 4. Assessment of Male Productive Function

The cornerstone of investigation of male fertility is semen analysis. However, routine semen analysis provides information only on concentration, motility and morphology of sperm rather than function [45]. More recently, further assessment tools have been developed. Damage to the DNA of sperm can be measured by various methods including sperm chromatin structure assay (SCSA), terminal deoxynucleotide transferase-mediated deoxyuridine triphosphate nick-end labelling (TUNEL), sperm chromatin dispersion (SCD) and COMET [46,47,48,49,50,51]. There are several clinically available techniques to measure levels of ROS, as a proposed mechanism by which sperm DNA is damaged, including chemi-luminescence, MiOXSYS and OxiSperm [5,7,9,52]. The sperm plasma membrane is especially susceptible to lipid peroxidation by ROS due to its high content of fatty acids. Lipid peroxidation can be assessed using the thiobarbituric acid reactive substances (TBARS) assay [53]. As a corollary, the total antioxidant capacity (TAC) can be measured by an assay to establish the cumulative effect of all antioxidants present in the sample [54]. There are, however, no current clinical recommendations to guide the use of these tests. The ability to assess the differential expression of proteins carried by human sperm is currently experimental only [55].

## 5. Elevated Seminal ROS and RPL

Kamkar et al. (2018) collected semen from 42 men with a history of spontaneous miscarriage and 42 fertile men as controls. Sperm DNA fragmentation (SDF) was measured using both the SCSA (sperm chromatic structure assay) and TUNEL methods [45]. The amount of sperm free radicals was measured using the luminescence method and a TAC kit was used for total antioxidant count (TAC). The amount of free radicals and the percentage of sperm DNA damage was significantly higher in the spontaneous miscarriage group than the control group. The TAC was lower in the spontaneous miscarriage group than the control group. Expanding this into recurrent miscarriage, Venkatesh et al. studied men from couples with a history of RPL and divided them into 2 groups: men with normal sperm parameters and men with abnormal sperm parameters [56]. They also used a control group of healthy men without a history of PRL. Sperm DNA damage was measured using the SCSA method. There was a positive correlation between sperm DNA damage and ROS with RPL. Using the SCSA method, Imam et al. (2011) compared the sperm of 20 men whose partners had a history of RPL with 20 healthy controls whose partners had no history of RPL [57]. They found significantly higher levels of both sperm DNA damage and ROS in the RPL group. TAC was lower in the RPL group compared with controls. A study by Jayasena et al. (2019) found that male partners of women with RPL had 4-fold increased levels of seminal ROS compared with the control group. 1/3 of all male partners of women affected by RPL had increased ROS compared with 10% of controls [58]. They also demonstrated greater levels of sperm DNA damage than controls [58]. Interestingly, in this study by Jayasena et al. higher levels of genitourinary infection or varicocele were not identified in men from couples affected by RPL [58]. A randomised controlled trial by Ghanaie et al. of couples affected by RPL and a varicocele in the male partner identified improved pregnancy and reduced miscarriage rates in men following varicocele repair compared with those whose varicocele was untreated [59]. This was supported by Negri et al. who reported miscarriage rates in line with the general population following repair of varicocele in patients with a prior history [60]. The study by Jayasena et al. was potentially limited by the relatively small sample size (n = 88) with which to assess this association [58].

The study by Jayasena et al. did not use men with proven fertility as controls. Potentially some of the controls may have latterly been found to be infertile, though arguably using men without proven fertility offers more robust results [58]. In contrast, a study by Gil-Villa et al. (2010), assessing sperm DNA damage, antioxidant capacity, lipid peroxidation and semen parameters in men from couples affected by RPL, used controls with proven fertility [53]. Gil-Villa reported increased TBARS and lower TAC in the RPL group compared with controls [53].

In contrast, Bellver et al., assessing sperm DNA damage and ROS in 3 groups of men, RPL vs. oligospermic men vs. healthy controls, found that whilst there was an association between elevated ROS and SDF, there was no relationship with RPL [61]. Table 1 summarises the methodologies of the aforementioned studies.

## 6. Sperm DNA Damage and RPL

An increased rate of sperm DNA damage has been identified in sporadic pregnancy loss, often in the setting of IVF or ICSI [18]. Attempts have been made in the literature to delineate the contribution of sperm DNA to the aetiology of RPL. Various groups have shown that the male partners of couples affected by RPL have greater rates of damage to the sperm DNA [48,62,63,64,65,66,67]. Failure to demonstrate this relationship has also been reported [53,67].

A 2019 systematic review including 15 studies on sperm DNA damage and RPL, of which 13 were included in a meta-analysis, found a significantly higher rate of sperm DNA damage in the male partners of women affected by RPL compared to the male partners of fertile control women [68]. It is important to note that the heterogeneity between studies limits the validity of the conclusions drawn (Table 2). For example, of the studies included in the meta-analysis, 8 enrolled participants after 3 miscarriages [48,57,63,67,69,70,71], 6 enrolled participants after 2 [62,64,65,72,73,74], and only in 5 did they have to be consecutive [57,64,67,73,74]. Only 4 studies excluded participants with concomitant infertility [64,65,70,73]. One study did not describe the work-up they performed to rule out alternative causes of RPL [69]; a normal uterus and negative antiphospholipid antibody testing in the female partner was required in the other studies. Two studies required a normal prolactin, 13 studies required normal thyroid function, 2 studies tested for diabetes, and 10 established the parental karyotype. Eight studies used frozen sperm whilst the remaining 7 used fresh sperm. There was variation in the techniques used to measure SDF: 6 studies used SCD test, 6 used TUNEL assay, 3 used SCSA, 1 used comet assay, 1 used acridine orange and 1 used aniline blue. Four studies used more than one test [48,62,67,70]. All studies performed standard semen analysis as well as testing for sperm DNA damage. Venkatesh et al. separated the men from couples affected by RPL by normal and abnormal semen analysis; both groups had significantly higher SDF compared with controls [56].

Absalam et al. studied 30 men from couples affected by RPL and 30 men who were not (control) in the setting of a fertility centre. Sperm DNA damage, measured via SCD, was higher in the affected group compared with controls. Importantly, the control group was also recruited from the infertility setting [69]. Bareh et al. performed a prospective cohort study comparing 26 men whose partners had a history of RPL with 31 controls with basic semen analysis and proven fertility [72]. Sperm DNA damage was measured using a TUNEL assay and found to be significantly higher in men with whose partners had a history of RPL compared with controls. Brahem et al. compared the semen samples of 31 men from couples with a history of RPL with a control of 20 men from couples who had both no history of RPL but also proven fertility [72]. Using a TUNEL assay they found significantly higher levels of DNA fragmentation (6.4%) compared with the control group (2.1%). This is similar to the findings by Carell et al. who, using a TUNEL assay, found significantly higher SDF in the RPL group (4.2%) compared with control (2.0%). Iman et al., Kumar et al., Ribas-Maynou et al., Ruizue et al., Zhang et al. and Zibi-Irab had similar findings, suggesting an association of sperm DNA damage and RPL [57,62,63,64,65]. Bhattacharya et al. compared 74 men whose partners had a history of RPL with 65 men of proven fertility [74]. They found significantly higher sperm DNA damage in the RPL group. Interestingly they did not find significant differences in sperm concentration and progressive motility, which is in contrast to 8/15 of the studies included in the systematic review and meta-analysis by McQueen et al. [68,74].

However, a finding of elevated levels of sperm DNA damage in men from couples affected by RPL has not been universally confirmed in the literature. Esquerre-Lamare et al. compared 33 men from couples affected by RPL with 27 controls [70]. The controls were recruited from a maternity unit both with recent proven fertility but also excluded if they had a history of ART or RPL [70]. Using both SCSA and TUNEL they found no statistical significance in sperm DNA damage between groups. The control group had significantly lower rates of abnormal sperm motility and morphology. Coughlan et al. (2015) compared the levels of sperm DNA damage in the male partners of 35 women with recurrent implantation failure following IVF vs. the male partners of 16 women with RPL vs. a control group of 7 recent fathers [67]. Sperm DNA damage was measured with both SCD and TUNEL methods [67]. There were no statistically significant differences between sperm concentration, morphology or motility between the three groups. Levels of sperm DNA damage in all groups was significantly lower in all groups when measured by SCD than when measured by TUNEL. Importantly, there was no significance between the affected and control groups when measured with either test. This should be interpreted with caution due to the small sample size of controls; indeed, this small sample size could account for the lack of significant difference in basic parameters between groups [67]. Gil-Villa et al. compared the semen samples of 23 men from couples with a history of RPL with 11 health men with proven fertility [53]. They assessed sperm DNA damage, antioxidant capacity, lipid peroxidation as well as basic semen parameters. The men in the control group had greater rates of normal basic sperm parameters and antioxidant capacity than men from couples affected by RPL. The RPL group had higher levels of lipid peroxidation and teratozoospermia than controls. However, there was no significance difference in sperm DNA damage between groups when measured using SCSA [53].

## 7. Sperm Protein Expression and RPL

The sperm proteome has been identified as providing a non-genomic contribution to embryo development [75,76,77]. Sperm proteins are able to undergo post-translational modification [55]. It is proposed that oxidative stress could induce post-translational modification of the spermatic proteins that are required for embryonic development such that they no longer function normally [55]. Mohantry et al. designed a study to investigate this hypothesis by establishing protein carbonylation and lipid peroxidation levels in couples affected by RPL [55]. The rationale for these measurements was as follows: high levels of lipid peroxidation would suggest ROS-induced damage to the sperm membrane, whilst the major contributors to protein carbonylation are the reactive carbonyl compounds generated during lipid peroxidation [78]. The RPL group comprised 16 men whose partners had suffered at least 2 miscarriages prior to 20 weeks of gestational age and who had no female factor to account for the losses. The control group comprised 20 men with proven fertility (live birth) within the last 12 months. The men were assessed by routine semen analysis, measurement of protein carbonylation via dinitrophenylhydralazine assay, and lipid peroxidation via TBARS. The study found a statistically significant correlation between lipid peroxidation and protein carbonylation, suggesting that lipid peroxidation contributes to carbonyl stress. Both lipid peroxidation and protein carbonylation negatively correlated with sperm count, motility and morphology. In this study TBARs were assessed and found, as in the study by Gil-Villa, to be elevated in the male partners of women affected by RPL [53,55]. The authors concluded that TBARs are a surrogate marker for cumulative oxidative damage to lipids, proteins and DNA in sperm.

Histones and protamines are basic proteins associated with the chromosome. During the later stages of spermatogenesis a histone-to-protamine transition occurs under epigenetic control [79]. This results in rearrangement and compaction within the sperm nuclei [80]. This exchange continues to occur after the sperm have left the testis. However 10–15% of the human genome remains associated with a histone [79]. These retained histones are more dense in areas associated with post-fertilisation events [79]. Abnormal histone retainment may play a role in pregnancy loss [55]. If histones are abnormally retained, specifically in areas protected from the normal spermatic post-fertilisation demethylation, embryonic development may become aberrant [76,77]. Studies measuring, via aniline blue staining, the degree of persistence of histones in the sperm nucleus have found greater rates of retained histones and thus aberrant sperm chromatin packaging in the RPL group compared with controls [55,63,64,81]. In a prospective study by Mohantry et al. (2020), standard semen parameters and protein expression in the male partners of women who had suffered RPL (defined as 2 or more losses) were compared with healthy controls [82]. The study found significantly different expression of 36 proteins in the RPL group compared to the control group [82]. This included under-expression of proteins within the spermatozoa known to protect the sperm from oxidative stress [82]. The same group published data suggesting that clusterin, an oxidative stress protein that also plays a role in post-fertilisation events, is underexpressed in the sperm of the male partners of women affected by RPL [83].

The seminal microenvironment offered by the fluid secreted by male accessory glands is considered to also potentially play a role in RPL [11]. The glands secrete cells carrying proteins and RNA into the fluid bathing the sperm. These proteins are hypothesised to regulate sperm maturation, histone removal and chromatic packaging, and thus ultimately spermatic function [11]. Furthermore, contact between the maternal endometrium and the proteins included in the male ejaculate are thought to aid decidualisation [11]. Decidualisation describes morphological and functional changes that occur within the endometrium in anticipation of implantation [11]. Jena et al. studied the secretory vesicles in human seminal fluids and found specific patterns of protein expression in the male partners of women experiencing recurrent pregnancy loss [11].

## 8. Use of Antioxidants

The clinical efficacy of antioxidant treatment for male infertility has been investigated in the literature and results have been conflicting [84,85,86]. Furthermore, the availability of over-the-counter antioxidants and their addition to various food products has raised concerns [87]. High levels of antioxidants can lead to reductive stress which is reported to be as damaging to cells as oxidative stress [87]. There is relatively little available data regarding the use of anti-oxidants in the RPL population. A study by Gil-Villa et al. (2009), administering antioxidant rich food or antioxidant supplementation to men with a history of PRL and increased DNA fragmentation index or TBARS, suggested promise in using antioxidant therapy to increase pregnancy rates [88]. In 2020, Nazari et al. enrolled 60 participants from the setting of recurrent miscarriage [89]. The men and their female partners were evaluated, with any known cause for RPL being an exclusion criterion. The female partners had normal anatomy and serum blood samples. Exclusion criteria in the male were age of 45, abnormal standard semen analysis, or a history of thrombophilia including antiphospholipid antibodies [89]. The period of abstinence prior to the sample was 3–7 days. They assessed basic semen parameters and sperm DNA damage via SCSA and TUNEL. Men were given antioxidant treatment, high in vitamin E and Zinc, for 90 days. Their post-treatment analysis showed an improvement in all sperm parameters. Sperm DNA fragmentation after treatment was statistically significantly lower than before treatment [89]. This study did not directly measure seminal ROS and there was no control group.

## 9. Conclusions

Evidence is mounting regarding an association between ROS and RPL, though this is not universally supported [58,61]. The exact underlying mechanisms by which elevated ROS may induce RPL has yet to be definitively elucidated but multiple lines of evidence suggest elevated ROS may cause structural and functional damage to the sperm [5,7,9,23,24,25,55]. This includes damage to the cell wall and mitochondria, DNA damage and the induction of epigenetic aberrations [5,7,9,23,24,25,55]. Systematic review has found significantly higher rates of sperm DNA damage in the male partners of women affected by RPL compared with unaffected men [68]. However, there is heterogeneity amongst the study designs which may account for the conflicting data. These include the change in definition of RPL, the range of methods used to assess seminal ROS and sperm DNA damage and the choice of controls. There is increased understanding that the sperm proteome and the seminal microenvironment may represent further potential targets for ROS-induced damage [55,71,79,82,83]. With 50% of cases of RPL being unexplained, research regarding the role of ROS in the aetiology of RPL is much needed [4]. This would allow the developing of targeted therapies for these affected couples. Moreover, further study is needed before recommendations regarding the use of antioxidants to reduce the rates of RPL can be made.

## Figures and Tables

**Figure 1 antioxidants-12-00723-f001:**
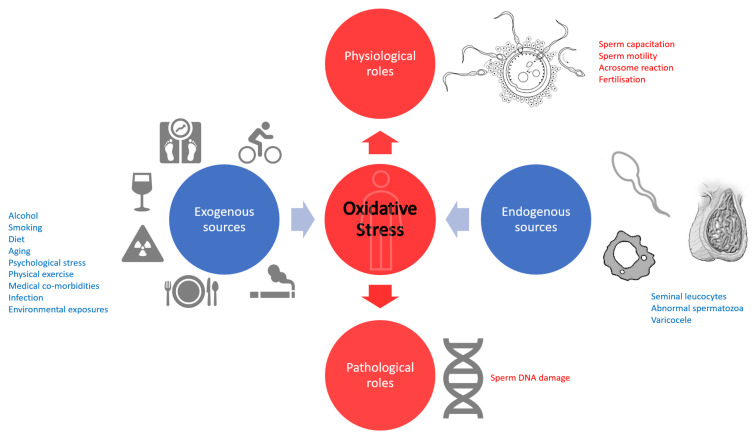
A schematic of example sources and roles of seminal oxidative stress.

**Table 1 antioxidants-12-00723-t001:** Qualitative analysis of studies on recurrent pregnancy loss, sperm DNA damage and reactive oxygen species.

Author	Year	Recurrent Pregnancy Loss Group		Control Group		Sperm Preparation	Sperm DNA Damage *	Oxidative Stress
		Definition	N=	Definition	N=			
Jayasena	2019	≥3 consecutive losses at <20/40	50	No co-morbidities	33	Fresh	SCD	Luminol-based chemiluminescence assay
Bellver	2015	≥3 losses at 5–14/40	30	Fertile + no history of RPL + normal karyotype + no co-morbidities	30	Fresh	SCD	Flow cytometric assay (OxyDNA) for sperm DNA oxidative damage
Imam	2011	≥3 consecutive losses at <20/40	20	≥1 live birth	20	Frozen	SCSA	Luminol-based chemiluminescence assay ELISA for total antioxidative capacity (TAC)
Venkatesh	2011	≥2 losses at <24/40	32	≥1 live birth	20	Frozen	SCSA	Luminol-based chemiluminescence assay

* SCSA: Sperm chromatic structure assay; SCD: Sperm chromatin dispersion.

**Table 2 antioxidants-12-00723-t002:** Qualitative analysis of studies on recurrent pregnancy loss and sperm DNA damage.

Author	Year	Recurrent Pregnancy Loss Group		Control Group		Sperm Preparation	Sperm DNA Damage *
		Definition	N=	Definition	N=		
Jayasena	2019	≥3 consecutive losses at <20/40	50	No co-morbidities	33	Fresh	SCD
Esquerre-Lamare	2018	≥3 losses at <12/40	33	≥1 live birth	27	Frozen with cryoprecipitate	SCSA and TUNEL
Zidi-Jrah	2016	≥2 losses at <24/40	22	≥1 live birth	20	Washed then frozen	TUNEL
Bareh	2016	≥2 losses at <20/40	26	≥1 live birth	31	Fresh	TUNEL
Bellver	2015	≥3 losses at 5–14/40	30	Fertile + no history of RPL + normal karyotype + no co-morbidities	30	Fresh	SCD
Coughlan	2014	≥3 consecutive losses at <20/40	16	≥1 live birth	7	Density centrifugation gradient vs. fresh	SCD and TUNEL
Ruixue	2013	≥3 losses at <12/40	68	Current pregnancy	63	Fresh	Aniline blue
Khadem	2012	≥3 losses at <20/40	30	Currently pregnancy	30	Fresh	SCD
Ribas-Maynou	2012	≥2 losses at <12/40	20	≥1 live birth	25	Frozen with cryoprecipitate	Comet and SCD
Kumar	2012	≥3 losses at <20	45	≥1 live birth	20	Frozen	SCSA
Zhang	2012	≥2 consecutive losses at <12/40	111	≥1 live birth	30	Fresh	SCD
Absalan	2012	≥3 losses at <20/40	30	Fertile + no history of RPL in partner	30	Fresh	SCD
Imam	2011	≥3 consecutive losses at <20/40	20	≥1 live birth	20	Frozen	SCSA
Venkatesh	2011	≥2 losses at <24/40	32	≥1 live birth	20	Frozen	SCSA
Brahem	2011	≥2 consecutive losses at <24/40	31	≥1 live birth	20	Frozen	TUNEL
Gil-Villa	2010	≥2 losses at <12/40	23	Established recent fertility	11	Fresh	SCSA
Bhattacharya	2008	≥2 consecutive losses at <8/40	74	≥1 live birth	65	Fresh	Acridine orange
Carrell	2003	≥3 losses at <20/40	21	≥1 live birth	26	Frozen	SCD and TUNEL

* SCSA: Sperm chromatic structure assay. SCD: Sperm chromatin dispersion. TUNEL: Terminal deoxynucleotide transferase- mediated deoxyuridine triphosphate nick-end labelling.

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
