# Peer review of "The Role of Seminal Oxidative Stress in Recurrent Pregnancy Loss"

_antioxidants, 2023, doi:10.3390/antiox12030723_

Round 1

Reviewer 1 Report

This manuscript is a revision on the effect of ROS and sperm DNA damage in recurrent pregnancy loss. The whole manuscript needs deep revision to provide a clear message on the relationship between ROS and pregnancy loss. The different sections just describe the results reported in the literature without giving any relevant clue or integrating the physiological relevance of the different approaches.

Moreover, the text contains several typographic and grammatical errors that must be corrected.

Author Response

Thank you for your feedback. Please find below an attached document detailing the changes made in response to your comments

With thanks 

Reviewer 2 Report

The Manuscript (2207245) entitled  The role of seminal reactive oxygen species and sperm DNA 2 damage in recurrent pregnancy loss” by Davies et al. discusses several important studies on the role of seminal reactive oxygen species and sperm DNA damages in recurrent pregnancy loss. The topic of this review is very important especially in clinical workup. An improved understanding of the causes of RLP may help to provide counselling and treatment targets to these patients.

Although, generally the manuscript is well written,  there are still some issues that need to be improved

My comments/suggestions are listed below.

Major issues:

1.     In the introduction section it would be valuable to add also other possible paternal factors that may play a role in RPL (genetic, epigenetic)

2.     Apart from articles discussed by Authors regarding semen oxidative stress and RLP several other have been also published which were not included in this review. In my opinion adding them to the manuscript provide a more thorough overview of the issue. Below examples of the articles:

-       Dhawan V, Kumar M, Deka D, Malhotra N, Singh N, Dadhwal V, Dada R. Paternal factors and embryonic development: Role in recurrent pregnancy loss. Andrologia. 2019 Feb;51(1):e13171. doi: 10.1111/and.13171. Epub 2018 Oct 15.

-       Mohanty G, Swain N, Goswami C, Kar S, Samanta L. Histone retention, protein carbonylation, and lipid peroxidation in spermatozoa: possible role in recurrent pregnancy loss. Syst Biol Reprod Med 2016;62:201-12

-       Mohanty G, Jena SR, Nayak J, Kar S, Samanta L . Proteomic Signatures in Spermatozoa Reveal the Role of Paternal Factors in Recurrent Pregnancy Loss. .World J Mens Health. 2020 Jan;38(1):103-114. doi: 10.5534/wjmh.190034. Epub 2019 Jul 3.

-       Mohanty G, Jena SR, Nayak J, Kar S, Samanta L. Quantitative proteomics decodes clusterin as a critical regulator of paternal factors responsible for impaired compensatory metabolic reprogramming in recurrent pregnancy loss.  Andrologia. 2020 Mar;52(2):e13498. doi: 10.1111/and.13498. Epub 2019 Dec 13.

-       Soumya R Jena, Jasmine Nayak, Sugandh Kumar, Sujata Kar, Anshuman DixitLuna Samanta. Paternal contributors in recurrent pregnancy loss: Cues from comparative proteome profiling of seminal extracellular vesicles. Mol Reprod Dev. 2021 Jan;88(1):96-112.

-       doi: 10.1002/mrd.23445. Epub 2020 Dec 20.

3.     I have some doubts concerning the paragraph “Use of antioxidants” as the studies discussed in it mainly concern use of antioxidants on basic semen parameters improvement and in some cases also on ART outcome . In my opinion Authors should focus here on studies related to topic of the review –RLP, oxidative stress and DNA damage, even if the number of such studies is not numerous.

Minor issues:

1.     change “traditional semen parameters” for “basic semen parameters”

2.     change the expression “men affected by RLP” (or similar) for “men whose partners  had experienced RLP”  or “male partners from couples experiencing RPL” or “male partners of women with RPL” (or similar)

3.     It would be more comprehensive if the table  would be divided on two separate tables: one with studies regarding sperm DNA damage and the second one with studies regarding oxidative stress

4.     Page 1,  lines 28-29 – at least the role of male factor in RLP is presented in ESHRE guidelines (from 2017 and the newest one from 2023)

https://www.eshre.eu/Guidelines-and-Legal/Guidelines/Recurrent-pregnancy-loss.aspx

5.     Page 1, line 42 – there should be “excess residual cytoplasm” – only it is a sperm anomaly - see articles below

-        Rengan AK, Agarwal A, van der Linde M, du Plessis SS.
An investigation of excess residual cytoplasm in human spermatozoa and its distinction from the cytoplasmic droplet.
Reprod Biol Endocrinol. 2012 Nov 17;10:92. doi: 10.1186/1477-7827-10-92.PMID: 23159014 Free PMC article. Review.

-        Cooper TG, Yeung CH, Fetic S, Sobhani A, Nieschlag E. Cytoplasmic droplets are normal structures of human sperm but are not well preserved by routine procedures for assessing sperm morphology. Hum Reprod. 2004 Oct;19(10):2283-8. doi: 10.1093/humrep/deh410. Epub 2004 Jul 8.PMID: 15242996

6.     Page 2, lines 45-48 – check spelling of “mitochondria”

7.     Page 2, line 48 – change “middle section of sperm” for “sperm midpiece”

8.     Page 2, line 56 – should be “intense physical activity” or “over-exercising”

9.     Page 3, line 100- Halosperm is another name for an assay for DNA fragmentation testing (sperm chromatin dispersion test)

https://www.halotechdna.com/productos/halosperm

10.  Page 5 , line 147 – Halosperm is a trade name for an in vitro diagnostic kit that allows the measurement of DNA fragmentation by SCD method (sperm chromatin dispersion)

11.  Page 7, line 268 – Change the phrase “The mean age of the participants was relatively old…” – the men can be old or young not “the mean age”

Author Response

(The authors gave the same response as above.)

Reviewer 3 Report

It is suggested to add : 1.To describe the machanism by which ROS causes DNA damage.2.To describe the sperm endogenious mechanisms that regulate and especially decreased ROS levels.

Author Response

(The authors gave the same response as above.)

Round 2

Reviewer 2 Report

I have no further comments to the revised verison of the menuscript.